# Dengue during pregnancy and live birth outcomes: a cohort of linked data from Brazil

Enny S Paixão,[1] Oona M Campbell,[1] Maria Gloria Teixeira,[2] Maria CN Costa,[2] Katie Harron,[3] Mauricio L Barreto,[4] Maira B Leal,[2] Marcia F Almeida,[5] Laura C Rodrigues[1]

¹IDE, London School of Hygiene & Tropical Medicine Faculty of Epidemiology and Population Health, London, UK
²Instituto de Saúde Coletiva, Federal University of Bahia, Salvador, Brazil
³Department of Health Services Research and Policy, London School of Hygiene & Tropical Medicine, London, UK
⁴Centro de Integração de Dados e Conhecimento para Saúde, Fiocruz, Salvador, Bahia, Brazil
⁵School of Public Health, Universidade de São Paulo, São Paulo, Brazil

**Correspondence to**
Dr Enny S Paixão;
npaixaoenfo@yahoo.com.br

## ABSTRACT

**Objectives** Dengue is the most common viral mosquito-borne disease, and women of reproductive age who live in or travel to endemic areas are at risk. Little is known about the effects of dengue during pregnancy on birth outcomes. The objective of this study is to examine the effect of maternal dengue severity on live birth outcomes.

**Design and setting** We conducted a population-based cohort study using routinely collected Brazilian data from 2006 to 2012.

**Participating** We linked birth registration records and dengue registration records to identify women with and without dengue during pregnancy. Using multinomial logistic regression and Firth method, we estimated risk and ORs for preterm birth (<37 weeks' gestation), low birth weight (<2500 g) and small for gestational age (<10thcentile). We also investigated the effect of time between the onset of the disease and each outcome.

**Results** We included 16 738 000 live births. Dengue haemorrhagic fever was associated with preterm birth (OR=2.4; 95% CI 1.3 to 4.4) and low birth weight (OR=2.1; 95% CI 1.1 to 4.0), but there was no evidence of effect for small for gestational age (OR=2.1; 95% CI 0.4 to 12.2). The magnitude of the effects was higher in the acute disease period.

**Conclusion** This study showed an increased risk of adverse birth outcomes in women with severe dengue during pregnancy. Medical intervention to mitigate maternal risk during severe acute dengue episodes may improve outcomes for infants born to exposed mothers.

## INTRODUCTION

Dengue is the most common viral vector-borne disease worldwide, with an estimated 390 million people infected each year, 96 million of whom develop clinical symptoms. It is endemic in over 100 countries (mostly in South and Central American and Southeast Asia) and is spreading to new areas; according to the WHO, approximately half of the world's population is at risk.[1] The effects of dengue during pregnancy are unclear, and there is a lack of evidence to inform women of reproductive age who live in or travel to endemic areas and who are at risk of dengue.

---

### Strengths and limitations of this study

► This is the first study to investigate the risk of small for gestational age infants born to women with dengue during pregnancy using a population-based approach with a sufficiently large sample size and controlling for confounders.
► Due to sample size, we were able to investigate rare exposure (such as dengue haemorrhagic fever).
► This study relies on the use of secondary data, and therefore identification of the outcome and the exposure is susceptible to misclassification.
► The linkage process was rigorously validated, and although we were unable to capture all cases, it is unlikely that the linkage process introduced bias, since linkage errors were not associated with any of our variables of interest.

---

The majority of studies published to date are cases series[2]; the seven studies published before 2016 with comparison groups show conflicting results,[3–9] and a meta-analysis published in 2016 found dengue during pregnancy was only associated with low birth weight and preterm birth in symptomatic women.[2] In 2017, a large Brazilian study found that notification of mild dengue during pregnancy was not associated with higher rates of preterm birth or lower birth weight in neonates compared with rates in a random sample of all neonates. However, this study did not investigate the effects of dengue by severity or date of disease onset.[10]

Maternal dengue, and specifically severe dengue, has been associated with fetal deaths,[11] but the effects on those who survive has not been investigated in population-based study. We analysed a large, population-based cohort to investigate the association between symptomatic dengue notified during pregnancy and adverse birth outcomes and to examine the effect of maternal dengue severity.

## METHODS

We performed a population-based cohort study by linking routine records of live births with records of women notified and confirmed with dengue disease in Brazil from 1 January 2006 to 31 December 2012. The methods described in this paper are similar to those used in a previously published paper about the effects of dengue on maternal deaths.[12]

### Data sources

We used the following data from two Brazilian databases.

1. The Live Births Information System (Sistema de Informação sobre Nascimentos; SINASC), which records live births in Brazil; these data are derived from the birth registration, a legal document completed by the health worker who attended the birth. It includes information on the mother (name, place of residence, age, marital status and education); the pregnancy (length of gestation and mode of delivery); and the neonate (birth weight and presence of congenital anomalies).[13] Data completeness is very high, with 97% of Brazilian births registered,[14] and most variables were >90% complete.[15]

2. Notifiable Diseases Information System (Sistema de Informação de Agravos de Notificação; SINAN), which records notifiable diseases. Dengue notifications include information on the individual (name, place of residence, age, sex and years of education), symptoms of the disease, laboratory tests and disease severity, assessed clinically and by laboratory according with the Brazilian dengue manual.[16] These data are reasonably complete on the variables used for linkage: <0.05% records were excluded because of a missing name. Around 5% of dengue cases did not have a final classification of severity. Laboratory confirmation was not required for notification or confirmation of dengue. During the study period (2006–2012), dengue was the main (and sometimes the only) vector-borne disease circulating in Brazil, as yellow fever and malaria occurred in restricted areas, and Zika and Chikungunya did not begin to circulate until 2014.

### Procedures

SINASC defines a live birth as the product of conception that, independent of the duration of pregnancy, breathes or shows any other signs of life (such as a heartbeat, umbilical cord pulsation or definite movement of voluntary muscles), after the separation from the mother's body.[13] We excluded live birth records from twin or other multiple pregnancies, congenital anomalies and records from municipalities without dengue notifications (figure 1).

### Outcomes

We assessed three birth outcomes: preterm birth (<37 weeks), low birth weight (<2500 g) and small for gestational age, calculated according to the Intergrowth Scale[17] using birth weight and gestational age in weeks at birth.

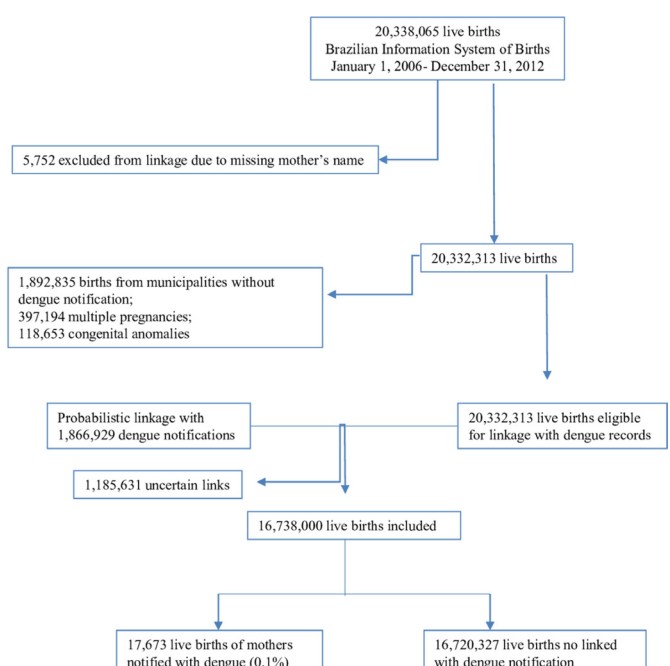

**Figure 1** Flow chart with the linkage information Brazil, 2006–2012.

This scale is sex specific, and those babies with a birth weight less than the 10th centile were considered small for gestational age. The data to calculate small for gestational age were available only for 2011 and 2012 (prior to this, gestational age was categorised).

### Exposure

We defined our exposure as confirmed cases of dengue notified to SINAN during a pregnancy that resulted in a live birth. In Brazil, confirmation of dengue is based either on clinical and epidemiological criteria (presence of clinical symptoms of dengue in the same area and time as other confirmed cases of dengue), or on clinical and laboratory criteria (the presence of clinical symptoms and a positive test for one of: (A) IgM detection by ELISA, (B) viral RNA detection via PCR, (C) NS1 viral antigen detection or (D) positive viral culture). We refer to laboratory confirmed cases as 'dengue during pregnancy, laboratory confirmed'. Dengue notifications do not always state whether the woman notified was pregnant; there is a field for this variable, but it is largely missing. We therefore defined maternal dengue cases by linking SINAN-confirmed dengue cases with live birth registrations within the 9 months following disease notification. To categorise disease severity, we used the clinical classification valid in Brazil in the period in which data were collected, which classified dengue into three clinical categories: (A) 'dengue fever', (B) 'complicated dengue' and (C) 'dengue haemorrhagic fever/dengue shock syndrome'. Dengue fever is a self-limiting disease, characterised as fever, with a severe headache, pain behind the eyes, muscle and joint pain and rash. Complicated dengue is a probable case of dengue characterised by one of the following: severe changes in the nervous system,

cardiorespiratory dysfunction, insufficient hepatic function, gastrointestinal bleeding, cavity spills or thrombocytopenia equal or less than 50 000/mm$^3$ or leucometry less than 1000/mm[18]; dengue haemorrhagic fever follows the WHO criteria and is characterised by fever, haemorrhagic evidence, thrombocytopenia and evidence of plasma leakage. Complicated dengue is a Brazilian definition for severe cases of dengue that do not meet WHO criteria for dengue haemorrhagic fever and cannot be classified as mild self-limited disease (dengue fever) due to their severity.

### Linkage process

We linked records of women with confirmed dengue (SINAN) with records of live births (SINASC) up to 9 months after the SINAN notification to identify those mothers who had dengue during pregnancy. Since there was no unique identifier for linkage between the two data sources, we relied on a set of identifiers available with good data quality in both datasets: name, age or date of birth and the place of residence of the mother at time of delivery/time of notification. We excluded records with missing or implausible names (eg, the name of a hospital as the name of the mother) (figure 1).

We used probabilistic linkage, which generates match weights that represent the likelihood that two records belong to the same individual based on the similarity between records from different datasets. This linkage method takes into account errors and missing values in identifiers and is useful when there is no unique identifier. Match weight calculations used the Fellegi-Sunter method.[19] For each record pair, we calculated a probabilistic match weight based on two conditional probabilities: m-probability, P (agreement|match) and u-probability, P (agreement|non-match). Match weights were calculated by summing the log of the ratio of m-probabilities and u-probabilities across different identifiers. The algorithm was implemented in Stata V.14.1 and R V.3.4.1.

Linkage followed the same procedure as previously described for linking stillbirth records and the dengue database in Brazil, in which we demonstrated a sensitivity (the proportion of true links captured) of 62% and a positive predictive value (the proportion of true matches among linked records) of 95%.[20] To estimate parameters for linkage weights and to validate the quality of the linkage, we created a gold standard dataset based on data from two states for 2010 Ceará and Espirito Santo). These states were chosen as they included a manageable number of records with which to conduct manual review different and represented a wide range of social economic status; 2010 was chosen since this was dengue epidemic year. Record pairs were ordered by match weight and manually reviewed to establish true matches (records belonging to the same mother) and to identify any erroneous links (false matches). Live birth records linked to confirmed dengue notifications with high certainty were classified as dengue in pregnancy cases, and birth records we were confident had no link were classified as non-dengue

cases. We excluded any records for which there was an uncertain link (where we could not establish whether a woman had dengue) to avoid misclassification (figure 1). Following linkage, the data were deidentified prior to analysis.

### Statistical analysis

We estimated the crude and adjusted risk ratios (RRs) and CIs from multinomial logistic regression. We compared live births from women without dengue during pregnancy (our reference group) with live births from women with notified dengue during pregnancy. We controlled for maternal age (categorised as ≤20; between 20–35 and ≥35 years old), education (less than 3 years; between 4 years and 7 years and more than 8 years) and marital status (single/widow/divorced or married/stable union), as these variables were also available in the data. To assess the sensitivity of results to the validity of clinical/epidemiological diagnosis, we repeated the analyses using laboratory-confirmed dengue cases only.

In general, dengue is an acute disease with rapid recovery. However, we investigated the effect of time between disease onset and live birth outcomes. The time between disease onset and birth outcome was calculated using the date of the disease onset (information available in SINAN) and the date when the outcome occurred (date of live birth); we categorised this difference as being less than or equal to 10 days or greater than 10 days.

We also evaluated the association between dengue and birth outcomes according to severity of disease. The exposure was classified as mild dengue, dengue with complications and dengue haemorrhagic fever. For this analysis, we calculated the OR using the Firth method (to reduce the small sample bias in maximum likelihood estimation) as we were analysing rare events and controlled for maternal age, education and marital status.[21]

### Patient and public involvement

There were no patients involved in the research. The data used are administrative routinely collected data.

## RESULTS

A total of 20 333 482 live births were recorded in SINASC between 2006 and 2012. After exclusions, 16 738 000 live births were included in the study, and 17 673 (0.1%) of their mothers were linked with a dengue notification record (figure 1).

The cohort characteristics by dengue status are described in table 1. Compared with pregnant women without dengue, pregnant women with dengue were more likely to have more years of formal education and to have delivered by caesarean section.

Maternal dengue was associated with a slight increase in the risk of preterm birth (from 7.3% to 7.9%, RR 1.1; 95% CI 1.0 to 1.2) and low birth weight (from 7.2% to 8.4%, RR 1.2; 95% CI 1.1 to 1.2), although the CI was borderline. There was no association between maternal

**Table 1** Maternal characteristics, delivery details and birth outcomes in relation to dengue status, Brazil, 2006–2012

| Characteristics | Notified with confirmed dengue in pregnancy n (%) | Without dengue notification n (%) |
|---|---|---|
| Age of the mother (years) | | |
| <20 | 4499 (25.5) | 4 384 159 (26.2) |
| 20–35 | 11 981 (67.8) | 11 042 342 (66.0) |
| >35 | 1193 (6.7) | 1 293 171 (7.8) |
| Missing | – | 655 (0.0) |
| Maternal education | | |
| Less than 3 years | 1065 (6.1) | 1 358 815 (8.3) |
| 4–7 years | 4661 (27.0) | 4 592 979 (28.1) |
| More than 8 years | 11 578 (66.9) | 10 399 703 (63.6) |
| Missing | 369 (2.1) | 368 830 (2.2) |
| Marital status | | |
| Single/widow/divorced | 11 518 (66.1) | 9 660 306 (58.7) |
| Married/union | 5899 (33.9) | 6 797 658 (41.3) |
| Missing | 256 (1.4) | 262 363 (1.6) |
| Numbers of prenatal visits | | |
| Inadequate (less than seven times) | 6991 (39.9) | 6 817 679 (41.2) |
| Adequate (seven or more times) | 10 515 (60.1) | 9 712 338 (58.8) |
| Missing | 167 (0.9) | 190 310 (1.1) |
| Delivery | | |
| Vaginal | 8332 (47.2) | 8 383 714 (50.2) |
| C-section | 9320 (52.8) | 8 307 072 (49.8) |
| Missing | 21 (0.1) | 29 541 (0.2) |
| Gestational age in the delivery | | |
| Less than 22 weeks | 17 (0.1) | 8844 (0.0) |
| 22–27 weeks | 68 (0.4) | 58 760 (0.4) |
| 28–31 weeks | 147 (0.8) | 112 407 (0.7) |
| 32–36 weeks | 1157 (6.7) | 1 011 633 (6.1) |
| Total preterm birth | 1389 (8.0) | 1 191 644 (7.2) |
| More than 37 weeks | 15 994 (92.0) | 15 215 440 (92.8) |
| Missing | 290 (1.6) | 313 243 (1.9) |
| Birth weight | | |
| ≥3000 | 12 171 (69.1) | 11 659 487 (69.9) |
| 3000–2500 | 3973 (22.5) | 3 816 201 (22.9) |
| Total low birth weight | 1484 (8.4) | 15 475 688 (7.2) |
| 1500–2500 | 1258 (7.1) | 1 026 248 (6.1) |
| <1500 | 226 (1.3) | 178 643 (1.1) |
| Missing | 38 (0.2) | 39 748 (0.2) |
| Small for gestational age (10th centile)* | | |
| Normal | 3681 (91.7) | 3 289 125 (92.0) |
| Small | 331 (8.2) | 287 173 (8.0) |
| Data not collected | 13 661 (77.3) | 13 144 029 (78.6) |

*Data available only in 2011 and 2012.
†% of each category without the missing value.

**Table 2**  Number of cases notified with dengue during pregnancy (N), crude and adjusted risk ratio for the association among dengue during pregnancy and preterm birth, low birth weight and small for gestational age

| | All dengue cases | | | Lab confirmed dengue cases | | |
|---|---|---|---|---|---|---|
| | N | Crude Risk Ratio (95% CI) | Adjusted Risk Ratio (95% CI) | | Crude Risk Ratio (95% CI) | Adjusted Risk Ratio (95% CI) |
| Preterm birth n=1 193 033 | | | | | | |
| Overall (<37 weeks) | 1389 | 1.1 (1.0 to 1.2) | 1.1 (1.0 to 1.2) | 439 | 1.1 (1.0 to 1.2) | 1.1 (1.0 to 1.2) |
| Less than 28 weeks | | 1.1 (1.0 to 1.1) | 1.1 (1.0 to 1.2) | | 1.0 (0.9 to 1.2) | 1.0 (0.9 to 1.2) |
| 28–32 weeks | | 1.2 (1.1 to 1.5) | 1.2 (1.1 to 1.4) | | 1.2 (0.9 to 1.6) | 1.2 (0.9 to 1.6) |
| 32–36 weeks | | 1.2 (1.0 to 1.5) | 1.1 (0.9 to 1.4) | | 1.1 (0.7 to 1.6) | 1.1 (0.7 to 1.6) |
| Low birth weight n=1 206 375 | | | | | | |
| Overall (<2500 g) | 1484 | 1.2 (1.1 to 1.2) | 1.2 (1.1 to 1.2) | 475 | 1.2 (1.0 to 1.3) | 1.2 (1.1 to 1.3) |
| 1500–2499 g | | 1.2 (1.1 to 1.2) | 1.2 (1.1 to 1.2) | | 1.2 (1.0 to 1.3) | 1.2 (1.1 to 1.3) |
| Less than 1500 g | | 1.2 (1.1 to 1.4) | 1.2 (1.0 to 1.3) | | 1.1 (0.9 to 1.4) | 1.1 (0.9 to 1.4) |
| Small for gestational age n=287 504 | | | | | | |
| <10th centile | 331 | 1.0 (0.9 to 1.1) | 1.0 (0.9 to 1.1) | 68 | 0.9 (0.7 to 1.1) | 0.9 (0.7 to 1.2) |

Brazil, 2006 –2012.
Adjusted for maternal age, education and marital status.

dengue and being small for gestational age (table 2). Restricting analyses to the 33%, (5755/17 388) of dengue cases that were laboratory confirmed did not change the magnitude of the associations (table 2).

Table 3 shows the dose–response effect of the severity of dengue on preterm birth, low birth weight and small for gestational age. Dengue haemorrhagic fever in pregnancy was associated with a doubling of the risk of preterm birth, from 1 191 644/16 407 084 (7.3%) to 12/79 (15.2%); OR 2.4 (95% CI 1.3 to 4.4) and an increase in the odds of low birth weight, from 1 204 891/16 680 579 (7.2%) to 11/79 (13.9%); OR 2.1 (95%

CI 1.1 to 4.0). We found no evidence for an effect of dengue on small for gestational age, even among pregnant women who developed haemorrhagic disease (287 173/3 576 298 (8.0%) versus 1/9 (11.1%); OR 2.5 (95% CI 0.4 to 12.2).

There was some evidence that the risk of dengue in pregnancy on birth outcomes depended on the time between dengue onset and the date of live birth: the magnitude of the effect of dengue on adverse birth outcomes was higher during the acute disease period, with some residual effects remaining after the first 10 days for preterm birth and low birth weight (table 4).

**Table 3**  Number of cases with dengue during pregnancy (N) and OR for the association dengue during pregnancy by severity of disease and adverse birth outcomes (preterm birth, low birth weight and small for gestational)

| Outcome | Dengue fever | Complicated dengue | Haemorrhagic fever |
|---|---|---|---|
| Preterm birth (<37 weeks) n=1 192 920 | | | |
| N (%) | 1234 | 30 | 12 |
| Crude OR (95% CI) | 1.1 (1.0 to 1.1) | 1.4 (0.9 to 1.9) | 2.4 (1.3 to 4.3) |
| Adjusted OR (95% CI) | 1.1 (1.0 to 1.1) | 1.4 (0.9 to 2.0) | 2.4 (1.3 to 4.4) |
| Low birth weight (<2500 g) n=1 206 265 | | | |
| N | 1327 | 36 | 11 |
| Crude OR (95% CI) | 1.1 (1.1 to 1.2) | 1.6 (1.2 to 2.3) | 2.1 (1.1 to 4.0) |
| Adjusted OR (95% CI) | 1.1 (1.1 to 1.2) | 1. 6 (1.1 to 2.3) | 2.1 (1.1 to 4.0) |
| Small for gestational age (10th centile) n=287 474 | | | |
| N | 294 | 6 | 1 |
| Crude OR (95% CI) | 1.0 (0.9 to 1.1) | 2.2 (0.9 to 5.1) | 2.0 (0.3 to 11.4) |
| Adjusted OR (95% CI) | 1.0 (0.9 to 1.1) | 2.3 (1.0 to 5.3) | 2.1 (0.4 to 12.2) |

Brazil, 2006–2012.
Adjusted for maternal age, education and marital status.

**Table 4** Number of cases with dengue during pregnancy (N) and risk ratio (RR) for the association dengue during pregnancy by timing of disease and adverse birth outcomes (preterm birth, low birth weight and small for gestational)

| Outcome | Outcomes within 10 days of disease onset (95% CI) | Outcomes after 10 days from disease onset (95% CI) |
|---|---|---|
| Preterm birth n=1 191 719 | | |
| N | 75 | 1314 |
| Crude RR (95% CI) | 2.1 (1.7 to 2.7) | 1.1 (1.0 to 1.1) |
| Adjusted RR (95% CI) | 2.0 (1.6 to 2.6) | 1.1 (1.0 to 1.1) |
| Low birth weight n=1 206 375 | | |
| N | 74 | 1410 |
| Crude RR (95% CI) | 2.1 (1.6 to 2.6) | 1.1 (1.1 to 1.2) |
| Adjusted RR (95% CI) | 2.0 (1.6 to 2.6) | 1.1 (1.1 to 1.2) |
| Small for gestational age (<10th centile) n=287 173 | | |
| N | 4 | 83 |
| Crude RR (95% CI) | 0.5 (0.2 to 1.5) | 1.0 (0.9 to 1.1) |
| Adjusted RR (95% CI) | 0.5 (0.2 to 1.5) | 1.0 (0.9 to 1.1) |

Brazil, 2006–2012.
Adjusted for maternal age, education and marital status.

## DISCUSSION

Our study demonstrates that dengue haemorrhagic fever in pregnancy is associated with a doubling of the risk of preterm birth (OR: 2.4) and of low birth weight (OR: 2.1) and that mild dengue fever is associated with an increase of 10%–20% in the risk of preterm birth and low birth weight, respectively. We found no evidence for an increased risk of small for gestational age for either severe or mild disease. The association between dengue and birth outcomes was strongest during the acute disease phase, within the first 10 days of disease onset.

Our results are consistent those with Nascimento et al,[10] who showed that symptomatic dengue infection did not greatly affect the risk of low birth weight (OR: 1.00, 95% CI 0.85 to 1.17) or preterm birth (OR: 0.98, 95% CI 0.83 to 1.16) when compared with a random sample of live births. However, we have previously shown that mild dengue is associated with a doubling of the risk of stillbirth (and that severe cases are associated with a fivefold increase in risk). By considering only live birth outcomes and excluding births with congenital anomalies (<1% of the sample) in this study, we may be underestimating the effect of dengue during pregnancy. Our findings are also consistent with a meta-analyses published in 2016,[2] which found an association between symptomatic dengue and preterm birth and low birth weight within in a hospital-based population that sought care and was likely to include only severe dengue cases. In our study, the association between maternal dengue and birth outcomes was strongest for women with dengue haemorrhagic fever in pregnancy.

There is evidence that different infectious diseases in pregnancy can lead to adverse outcomes. The evidence is still fragmented, and there is no consistent effect, which appears to vary according to the pathogen involved and the timing and severity of maternal disease. For instance, studies comparing pregnant women with infection to pregnant women without found: (A) severe influenza increased the risk of preterm birth from 2.4 to 4 times, whereas studies based on mild range of illness did not find an effect[22] and (b) measles and HIV infection increased the risk of low birth weight by 3.5 and 1.6 times, respectively.[23 24]

One mechanism by which dengue during pregnancy affects preterm birth and low birth weight that is consistent with our results is through maternal illness, rather than a direct effect on the fetus. Since we did not observe an effect on small for gestational age, the effect of dengue in pregnancy on birth weight may have occurred via prematurity of the newborn. This assumption is supported by the higher risk of preterm births observed among mothers within less than 10 days after the onset of the disease and among those with complicated disease and haemorrhagic dengue. The higher risk of preterm birth could be due to the early onset of labour delivery or by early delivery due to medical interventions, such as caesarean section, required because of concern about risk to the mother. However, data on small for gestational age was only available for 2011–2012, which may have affected our ability to observe an effect. Furthermore, analysis according to mode of delivery is complicated due to high rates of caesarean section in Brazil, many of which are not clinically indicated.

Limitations of this study include the linkage process and use of secondary data. First, outcome and exposure identification is susceptible to misclassification using secondary sources. Preterm birth and small for gestational age are underascertained in routinely collected data from low-income and middle-income countries. However, as this is likely to have affected both those

exposed and not exposed to dengue in a similar way, it is unlikely that results of this study would have been biased.[25 26] Regarding the dengue diagnosis, it is usual practice in outbreaks/epidemics to test until the origin of the outbreak is clearly established and after that only test when there is a clinical indication. The proportion of laboratory confirmed cases among general population in Brazil is around 30%,[27] and therefore, misclassification based on the lack of laboratory confirmation is possible. However, our sensitivity analysis demonstrated similar results in subsets of the data with and without laboratory confirmation of dengue. Second, our linkage was rigorously validated in a previous study.[19] Despite the low sensitivity (that kept us from making statements about the prevalence of dengue during pregnancy), it is unlikely that the linkage process introduced bias, since linkage errors occurred randomly within the data and were not associated with our variables of interest. Finally, although we adjusted for confounders, other possible unknown confounders, such as maternal comorbidities or quality or type of obstetric care, may have contributed to the association between severe dengue and adverse birth outcomes.

As dengue prevention is not straightforward and mosquito control has failed in Brazil, our findings highlight the crucial role of the clinical management of dengue in pregnant women. Doctors and medical staff in general attending to pregnant women with dengue should closely observe and monitor the patients to be able to intervene timely and avoid adverse outcomes for mothers and babies, especially during the acute phase. We recommend careful observation and notation of dengue in antenatal records and further research into the association between dengue and adverse pregnancy outcomes.

In summary, this study demonstrates a doubling in the risk of preterm birth and low birth weight associated with dengue haemorrhagic fever in pregnancy and increases of 10%–20% in women with mild dengue. One contributing factor to adverse birth outcomes may be medical interventions triggered to mitigate mothers risk associated with dengue haemorrhagic fever. We recommend further research in different settings to confirm our results and additional studies of adverse birth outcomes for other vector-borne diseases.

**Contributors** ESP, KH and MBL carried out the analysis. ESP wrote the first draft of the article. LCR and MGT conceived the study. MCNC, MLB, OMC and MFA contributed to the study design and interpretation. All authors revised the manuscript and approved the final version.

**Funding** ESP was funded by National Council for Scientific and Technological Development (CNPq-Brazil); LR is partially funded by the European Union's Horizon 2020 research and innovation program under Zika-PLAN grant agreement No. 734584; KH is funded by the Wellcome Trust (grant number 103975/Z/14/Z).

**Disclaimer** The funder of this study had no role in study design, data collection, data analysis, data interpretation or writing of the report.

**Competing interests** None declared.

**Patient consent for publication** Not required.

**Ethics approval** Ethical approval was obtained from The Federal University of Bahia, Salvador, Brazil (CAAE: 26797814.7.0000.5030) and from The London School of Hygiene & Tropical Medicine (Ethics Ref:10269).

**Provenance and peer review** Not commissioned; externally peer reviewed.

**Data sharing statement** The data used in this study are identified, therefore confidential, because they include patient personal information that can be traced back to individual. They are obtainable in the Brazilian Ministry of Health, but restrictions apply to the availability of these data, which were used under licence for the current study, and so are not publicly accessible. Data are however available from the Brazilian Ministry of Health on reasonable request for researchers who meet the criteria for access to confidential data, approved by the Ethics Committee. Email of a contact point in the Department of Health Information Ministry of Health Brazil: datasus@saude.gov.br.

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
