## [Reviewer comments · BMJ Open]

ARTICLE DETAILS

TITLE (PROVISIONAL)	Dengue during pregnancy and live birth outcomes: a cohort of linked data from Brazil
AUTHORS	Paixão, Enny; Campbell, Oona; Teixeira, Maria; Costa, Maria Conceição; Harron, Katie; Barreto, Mauricio; Leal, Maira; Almeida, Marcia; Rodrigues, Laura

VERSION 1 - REVIEW

REVIEWER	Katherine Ahrens US Department of Health and Human Services, Office of Population Affairs United States
REVIEW RETURNED	29-Jun-2018

GENERAL COMMENTS	BMJ Open Review by Katherine Ahrens, Office of Population Affairs, U.S. Department of Health and Human Services 06/29/2018 Dengue during pregnancy and adverse birth outcomes: a cohort analysis using routine data Brief synopsis: The authors linked 2006-2012 birth record data in Brazil to a notifiable disease system database to identify births to women with reported dengue infection in the previous 9 months. They used multinomial logistic regression to estimate adjusted risk ratios for preterm birth, low birthweight, and small-for-gestational age, by maternal dengue lab confirmation (all cases, lab-confirmed only), severity of infection (dengue fever [least severe], complicated dengue, dengue hemorrhagic fever [most severe]), and onset of dengue symptoms in relation to time of birth (within 10 days of birth, more than 10 days before birth), all in comparison to births that clearly didn't link to the notifiable disease system database. They found maternal dengue hemorrhagic fever doubled the risk of preterm birth and low birthweight, but did not have an effect on small-for-gestational-age birth. Births to mothers with onset of symptoms within 10 days of the birth were at higher risk for being preterm and low birthweight. One explanation for these findings might be that women with severe dengue infection are delivered early due to medical intervention. Overall comments: This is a clearly written manuscript on an important topic. I have some questions and concerns about the methodology, but overall the analysis and conclusions seems valid.
--

Major comments:

1. Can the authors provide a more compelling rationale for undertaking this study? In the introduction they cite a 2016 meta-analysis that found dengue during pregnancy was associated with low birthweight and preterm birth in symptomatic women, which is similar to the findings of the current study. What were the major limitations of that study and how could this new study overcome those limitations?
2. The authors mention the potential bias introduced by excluding stillbirths from their analysis, but what about the effect of excluding infants with congenital anomalies? On a related note, please include a brief summary in the Discussion of the findings from their previous paper on dengue and stillbirth- what is the magnitude of association that they found?
3. Can the authors speculate as to the clinical implications of their findings? How can the findings from their study be used to improve birth outcomes among dengue-infected women in Brazil?
4. Expand a little on the linkage procedures. Briefly explain the Fellegi-Sunter method and what "match weights" are. Explain why name, age or date of birth, and place of residence were used as linking variables. How many records were reviewed by hand for the "gold standard" dataset and how can the authors be sure this manual review was superior to the computer-assisted linkage? How certain are the authors that women who didn't match truly didn't have dengue during pregnancy? What software and linkage packages were used?
5. If the authors speculate that women with severe dengue disease during pregnancy are delivered early by medical intervention, then why didn't they look at c-section rates by dengue severity? Could they have used the birth certificate variables to distinguish spontaneous preterm birth vs. medically indicated preterm birth?
6. Is it expected that maternal dengue cases would represent more educated women? Does that indicate that higher socio-economic status women might be more likely to be tested for dengue and put in the notifiable disease system in Brazil? How would that affect the study results?

Minor comments:

1. Why did the authors adjust for maternal age, education, and marital status? Why not adjust for other potential confounders like socio-economic status or place of residence?
2. Why include low birthweight as an outcome? As the authors point out, this outcome is a mix of "born too early" and "born too small", so why bother including it as an outcome?
3. Table 1: add a row for total preterm birth and total low birthweight (if the authors keep this outcome). Put total sample size in first row (e.g., 17,673 maternal dengue cases). Put the asterisk in the table where it should go.
4. Tables 2, 3, and 4: Put total sample size in first row, and please take out the total percentages (e.g., "0.103%" for preterm birth and mild dengue in Table 2). Those percentages are completely meaningless. If the authors want to put in percentages, put in the percentage of preterm births among women with each of the dengue outcomes (e.g. 1,234 out of xxxxx total "mild dengue" cases equals xx%). For Table 2, label dengue disease severity the same way as described in text (text doesn't use term "mild dengue").
5. Isn't it possible that the maternal dengue cases could have included a women who had the disease 9 months ago and then became pregnant 8 months ago, delivering a 36 week old fetus?

	This would mean she didn't actually have dengue during her pregnancy. Perhaps the authors should only include maternal dengue cases that fell within the gestational age time frame recorded on the birth certificate.
--	--

REVIEWER	Suranjith L Seneviratne Institute of Immunity and Transplantation, Royal Free Hospital, London, UK Department of Surgery, Faculty of Medicine, University of Colombo, Sri Lanka
REVIEW RETURNED	15-Jul-2018

GENERAL COMMENTS	The authors have done a good analysis of the available data. I would like some studies on dengue during pregnancy from South Asia and South East Asia to be cited and included in the reference list.
---

REVIEWER	Dr B.V.K.M Bopeththa Registrar in Emergency Medicine Accident and Emergency department Teaching Hospital Kandy Sri Lanka
REVIEW RETURNED	04-Nov-2018

GENERAL COMMENTS	This is an interesting study which has shown the perinatal effects of dengue infection as there is very little evidence available on obstetric and neonatal complications of dengue. However, I would like to emphasize the following points. 1. There are four circulating sero types of dengue virus. Clinical manifestations and severity of disease of each virus are usually different. Therefore, it would be more informative, if you could provide further details of circulating viruses in your country or region. 2. The linkage process is based on multiple assumptions. The reproducibility of this process is questionable. Therefore further statistical expert opinion needs to be obtained prior to publication. 3. There are number of risk factors for the preterm delivery, low birth weight and small for gestational age. Dengue itself may not be the causative factor of low birth weight and pre term delivery. Therefore, discussion has to be expanded on these.
---

REVIEWER	Alessio Andronico Institut Pasteur
REVIEW RETURNED	15-Nov-2018

GENERAL COMMENTS	This paper focus on the effect of mild to severe dengue infections on birth outcomes through a cohort study based on data collected in Brazil from 2006 to 2012. The authors find a doubled risk of preterm birth and low birth weight associated to dengue hemorrhagic fever during pregnancy. They similarly observe an increased risk, though borderline significant, in the case of mild and complicated dengue infections during pregnancy, but they find no association between maternal infection and smaller size than normal for gestational age of newborns. The effect of dengue infection on birth outcomes was more clearly evidenced during the acute phase of the disease. The authors conclude that the
---

	observed birth outcomes may be explained by medical interventions (e.g. caesarean section) put in place to safeguard the mother. I found the article interesting to read, technically sound, and clearly written. I have only few marginal comments/pointers to typos:  1. Page 1 line 32: I think something got cut off from ‘...and linked to identify...’. 2. Page 1 line 33: ‘16,738,00’ should be 16,738,000. However, the sentence is then also repeated at line 38. 3. Page 1 line 43-44: I would rephrase the sentence ‘It appears that one contributing...’, which I found confusing. 4. Page 5 line 109: ‘(2006-2016)’ should be (2006-2012). 5. Page 6 line 123: there is a closing parenthesis after ‘age’. 6. The acronyms CI, OR, and RR are never defined in the text.
--	---

REVIEWER	Natalie Dean University of Florida
REVIEW RETURNED	27-Nov-2018

GENERAL COMMENTS	The authors conduct a population-based analysis using national registry data from Brazil to evaluate the relationship between exposure to dengue during pregnancy and adverse birth outcomes. A strength of this paper is the large sample size of over 16 million live births. Unfortunately, I note several weaknesses in the analytical approach used. Major: (1) I believe that the analyses by timing of disease are not meaningful as presented. The analysis uses the unnatural time scale of since dengue, as this was most readily calculable from the data, but this has important limitations. Consider the following examples:  o A pregnant woman infected in the first trimester has a 0% chance of having a *live birth outcome* within 10 days of onset. o A pregnant woman infected in the 38th week of gestation has a 0% chance of having a *pre-term birth.* Time since symptom onset is not meaningful because symptom onset can occur any time during pregnancy, but live outcomes primarily occur in the third trimester. Thus, I am not sure how to interpret the results in Table 4. Similarly, I am not sure that the sentence in the abstract “The magnitude of the effects was higher in the acute disease period” is meaningful. The most meaningful time scale is gestational age because that reflects the underlying biology of the process. The authors are strongly encouraged to restructure this portion of the analysis to a gestational age time scale. What is the risk of preterm birth for mothers with symptom onset during first, second, third trimester, again noting potential for bias due to missing fetal deaths. (2) There is a lot of potential for residual confounding here, obviously. The authors should consider further adjustments, such as adjusting for geographical region in a mixed effects analysis or some similar approach. (3) The authors posit that medical interventions (like c-section) explain some of the increased risk for women with dengue haemorrhagic fever, even including this hypothesis in their abstract. But they provide no support for this hypothesis (references, data). The data set includes data on mode of delivery. Why is this not examined?
---

	(4) Page 15, line 299, line 308: Non-differential misclassification leads to bias towards the null. Minor: (5) The text makes causal interpretations that are too strong for the analysis, e.g. p2, line 38: Dengue haemorrhagic fever doubled the risk of preterm birth. Should be tempered using words like "is associated with". (6) Numerous grammatical and typographical errors throughout, e.g. p2, line 33: It was included 16,738,000 births. (7) Table 2 caption reads "adjusted for the association among dengue during pregnancy and..." This confusingly makes it sound like you are adjusting for dengue instead of confounders. (8) Page 7, line 161: Define sensitivity in the context of matching records. (9) Table 1: With this sample size, everything will be statistically significant. It is better to talk in terms of meaningful differences (e.g. % differences).
--	---

REVIEWER	Mohammadreza Mohebbi Deakin University, Australia
REVIEW RETURNED	28-Nov-2018

GENERAL COMMENTS	The title can be more informative by including 'data linkage' and perhaps the fact that only live birth were considered. tables 2 and 4 shows risk ratio estimates, but the estimation methods has not explained in the methods. Table 3 has 3 strata for the outcome (Mild dengue, Complicated dengue, and Haemorrhagic fever). Is this the table that multinomial logistic regression with more than two strata for the outcome was implemented? Please clarify this in the methods. In multinomial logistic models usually estimated ORs presented according to a reference group. Is reference group non-dengue participants? please clarify. Has regions (states) heterogeneity effect been tested in the models? Were data on Urban/Rural place of residence extracted? Results section: 'Table 3 shows the dose-response effect of the severity of dengue on preterm birth, low birth weight and small for gestational age.' The presented model has only considered these factors as nominal. It would be informative to implement an official test for linear trend on these factors to support the argument. Flowchart with the linkage information is not self-explanatory. Please add explanation about 'probabilistic linkage' and 'uncertain links'. Figure 1 does not look like a ready for publication figure. Please redo this in a professional software. validation results of the quality of the linkage can be explained in further details. Why Ceará and Espirito Santo States for 2010 were used for linkage validation? Please justify.
--

	probabilistic linkage method needs more explanation. English and grammar needs editing.
--	--

VERSION 1 – AUTHOR RESPONSE

Reviewer 1	
Can the authors provide a more compelling rationale for undertaking this study? In the introduction they cite a 2016 meta-analysis that found dengue during pregnancy was associated with low birthweight and preterm birth in symptomatic women, which is similar to the findings of the current study. What were the major limitations of that study and how could this new study overcome those limitations?	Thank you for your comments. We have added the limitations of the previous studies and emphasised what this study adds to the literature.
The authors mention the potential bias introduced by excluding stillbirths from their analysis, but what about the effect of excluding infants with congenital anomalies? On a related note, please include a brief summary in the Discussion of the findings from their previous paper on dengue and stillbirth- what is the magnitude of association that they found?	We added the results of our previous study into the discussion, and have now included a discussion of the limitations of excluding congenital anomalies. We excluded this group because it is associated with preterm and low birth weight.
Can the authors speculate as to the clinical implications of their findings? How can the findings from their study be used to improve birth outcomes among dengue-infected women in Brazil?	We added the paragraph: “As dengue prevention is not straightforward and mosquito control has failed in Brazil, our findings highlight the crucial role of the clinical management of dengue in pregnant women. Doctors and medical staff in general attending to pregnant women with dengue should closely observe and monitor the patients to be able to intervene timely and avoid adverse outcomes for mothers and babies, especially during the acute phase. We recommend careful observation and notation of dengue in antenatal records and further research into the association between dengue and adverse pregnancy outcomes.”
Expand a little on the linkage procedures. Briefly explain the Fellegi-Sunter method and what “match weights” are. Explain why name, age or date of birth, and place of residence were used as linking variables. How many records were reviewed	We have added information on the linkage procedures within the methods section. I can also include supplementary material with more details on the linkage (including numbers).

by hand for the “gold standard” dataset and how can the authors be sure this manual review was superior to the computer-assisted linkage? How certain are the authors that women who didn’t match truly didn’t have dengue during pregnancy? What software and linkage packages were used?	
If the authors speculate that women with severe dengue disease during pregnancy are delivered early by medical intervention, then why didn’t they look at c-section rates by dengue severity? Could they have used the birth certificate variables to distinguish spontaneous preterm birth vs. medically indicated preterm birth?	This is a very good and important question. We analysed the C-section rates by dengue status and did not observe difference, however it is important to understand the Brazilian context of C-section rates. Brazil has one of the highest rate of C-section worldwide, most of them occur without clinical reason, therefore, this can explain why we did not find difference by mode of delivery. We cannot distinguish between spontaneous and indicated preterm birth using the available data.
Is it expected that maternal dengue cases would represent more educated women? Does that indicate that higher socio-economic status women might be more likely to be tested for dengue and put in the notifiable disease system in Brazil? How would that affect the study results?	No, it is expected that less educated women might be more likely to be notified with dengue. However, incomplete records could be related to social disadvantage and consequently these records were less likely to match. If this is true our analyses might have underestimated the magnitude of the association between dengue and adverse fetal outcomes.
Why did the authors adjust for maternal age, education, and marital status? Why not adjust for other potential confounders like socio-economic status or place of residence?	We did not have others variables available in the dataset to adjust for. We only have municipality of residence that is not useful to discriminate social economic status in Brazil due to with high inequality within the same territory.
Why include low birthweight as an outcome? As the authors point out, this outcome is a mix of “born too early” and “born too small”, so why bother including it as an outcome?	We did not have gestational age for every year. This outcome is still an important outcome that is relevant to future health of the child
Table 1: add a row for total preterm birth and total low birthweight (if the authors keep this outcome). Put total sample size in first row (e.g., 17,673 maternal dengue cases). Put the asterisk in the table where it should go.	Done
Tables 2, 3, and 4: Put total sample size in first row, and please take out the total percentages (e.g., “0.103%” for preterm birth and mild dengue in Table 2). Those percentages are completely meaningless. If	Done

the authors want to put in percentages, put in the percentage of preterm births among women with each of the dengue outcomes (e.g. 1,234 out of xxxxx total “mild dengue” cases equals xx%). For Table 2, label dengue disease severity the same way as described in text (text doesn’t use term “mild dengue”).	
Isn't it possible that the maternal dengue cases could have included a women who had the disease 9 months ago and then became pregnant 8 months ago, delivering a 36 week old fetus? This would mean she didn't actually have dengue during her pregnancy. Perhaps the authors should only include maternal dengue cases that fell within the gestational age time frame recorded on the birth certificate.	We did use the gestational age time frame recorded on the birth certificate as suggested. There is a certain degree of uncertainty because this variable was only available in a categorized form.
Reviewer 2	
The authors have done a good analysis of the available data. I would like some studies on dengue during pregnancy from South Asia and South East Asia to be cited and included in the reference list.	Thank you.
Reviewer 3	
There are four circulating sero types of dengue virus. Clinical manifestations and severity of disease of each virus are usually different. Therefore, it would be more informative, if you could provide further details of circulating viruses in your country or region.	Thank you very much. We agree that information on dengue serotype would be informative, however we do not have this information available. The number of virus isolation is very small.
The linkage process is based on multiple assumptions. The reproducibility of this process is questionable. Therefore further statistical expert opinion needs to be obtained prior to publication.	We agree that there are a number of assumptions underpinning the linkage. Our previous peer-reviewed papers have addressed these issues in more detail. In this paper, we concentrate on the analysis of the linked data.
There are number of risk factors for the preterm delivery, low birth weight and small for gestational age. Dengue itself may not be the causative factor of low birth weight and	We agree that we have to interpret our results with caution to not imply causality.

pre term delivery. Therefore, discussion has to be expanded on these.	
Reviewer 4	
Page 1 line 32: I think something got cut off from ‘...and linked to identify...’. 2. Page 1 line 33: ‘16,738,00’ should be 16,738,000. However, the sentence is then also repeated at line 38. 3. Page 1 line 43-44: I would rephrase the sentence ‘It appears that one contributing...’, which I found confusing. 4. Page 5 line 109: ‘(2006-2016)’ should be (2006-2012). 5. Page 6 line 123: there is a closing parenthesis after ‘age’. 6. The acronyms CI, OR, and RR are never defined in the text.	Thank you very much. We made all these corrections.
Reviewer 5	
I believe that the analyses by timing of disease are not meaningful as presented. The analysis uses the unnatural time scale of since dengue, as this was most readily calculable from the data, but this has important limitations. Consider the following examples: 1. A pregnant woman infected in the first trimester has a 0% chance of having a *live birth outcome* within 10 days of onset. 2. A pregnant woman infected in the 38th week of gestation has a 0% chance of having a *pre-term birth.* Time since symptom onset is not meaningful because symptom onset can occur any time during pregnancy, but live outcomes primarily occur in the third trimester. Thus, I am not sure how to interpret the results in Table 4. Similarly, I am not sure that the sentence in the abstract “The magnitude of the effects was higher in the acute disease period” is meaningful. The most meaningful time scale is gestational age because that reflects the underlying biology of the process. The authors are strongly encouraged to restructure this portion of the analysis to a gestational age time scale. What is the risk of preterm birth for mothers with symptom onset during first, second, third trimester, again noting potential for bias due to missing fetal deaths.	Thank you very much for your valuable inputs. We understand your point of view, however our data on gestational age was aggregated, therefore we could not measure the trimester without a degree of uncertainty. We decided to not present this information, but have included this within the discussion. We think that in combination with the analysis by severity of disease, these findings still provide a case for supporting the argument that adverse outcomes are related with acute illness.

There is a lot of potential for residual confounding here, obviously. The authors should consider further adjustments, such as adjusting for geographical region in a mixed effects analysis or some similar approach	We do not have these data.
The authors posit that medical interventions (like c-section) explain some of the increased risk for women with dengue haemorrhagic fever, even including this hypothesis in their abstract. But they provide no support for this hypothesis (references, data). The data set includes data on mode of delivery. Why is this not examined?	This is a very good and important question. We analysed the C-section rates by dengue status and did not observe difference, however it is important to understand the Brazilian context of C-section rates. Brazil has one of the highest rate of C-section worldwide, most of them occurs without clinical reason, therefore, this can explain why we did not find difference by mode of delivery.
The text makes causal interpretations that are too strong for the analysis, e.g. p2, line 38: Dengue haemorrhagic fever doubled the risk of preterm birth. Should be tempered using words like "is associated with".	We made this change.
Numerous grammatical and typographical errors throughout, e.g. p2, line 33: It was included 16,738,000 births.	Thank you! We revised the manuscript.
Table 2 caption reads "adjusted for the association among dengue during pregnancy and...." This confusingly makes it sound like you are adjusting for dengue instead of confounders.	Done
Page 7, line 161: Define sensitivity in the context of matching records	Done
Table 1: With this sample size, everything will be statistically significant. It is better to talk in terms of meaningful differences (e.g. % differences).	We agree and excluded the p value.
Reviewer 6	
The title can be more informative by including 'data linkage' and perhaps the fact that only live birth were considered.	Thank you. The new title is: "Dengue during pregnancy and live birth outcomes: a cohort of linked data from Brazil
tables 2 and 4 shows risk ratio estimates, but the estimation methods has not explained in the methods.	We include more information to clarify how we estimated the risk ratio using multinomial logistic regression and the reference group (women without dengue).
Table 3 has 3 strata for the outcome (Mild dengue, Complicated dengue, and Haemorrhagic fever). Is this the table that multinomial logistic regression with more than two strata for the outcome was implemented? Please clarify this in the methods.	No, in this table we used the Firth method to calculate. We clarified this in the method.

In multinomial logistic models usually estimated ORs presented according to a reference group. Is reference group non-dengue participants? please clarify	Yes, the reference group is women without dengue during pregnancy, we added this information in the methods.
Has regions (states) heterogeneity effect been tested in the models?	No, this was not the objective of the study.
Were data on Urban/Rural place of residence extracted?	No, we did not have these data available.
'Table 3 shows the dose-response effect of the severity of dengue on preterm birth, low birth weight and small for gestational age.' The presented model has only considered these factors as nominal. It would be informative to implement an official test for linear trend on these factors to support the argument.	In the individual data these information is only categorical. Three categories do not represent a linear scale.
Flowchart with the linkage information is not self-explanatory. Please add explanation about 'probabilistic linkage' and 'uncertain links'.	We have added further explanation in the main text.
Figure 1 does not look like a ready for publication figure. Please redo this in a professional software.	Thank you, we will work together with the journal to format the figure better.
validation results of the quality of the linkage can be explained in further details. Why Ceará and Espírito Santo States for 2010 were used for linkage validation? Please justify.	We used these two municipalities because: 1. We were not able to do a validation study with the all Brazilian data, the sample is too big; 2. 2010 was an year with dengue epidemic (better dengue vigilance); 3. these two municipalities are very different in social economic status therefore we compare the results between them to be certain that the linkage worked in the all country. We have a document explain all the details of the linkage validation that is going as supplement material.

VERSION 2 – REVIEW

REVIEWER	Natalie Dean Department of Biostatistics, University of Florida, USA
REVIEW RETURNED	11-Feb-2019

GENERAL COMMENTS	The authors have adequately addressed my comments. I have no further suggestions.
---

REVIEWER	Mohammadreza Mohebbi Deakin University
REVIEW RETURNED	13-Feb-2019

GENERAL COMMENTS	The reviewer completed the checklist but made no further comments.
--